# Breastfeeding Duration and High Blood Pressure in Children and Adolescents: Results from a Cross-Sectional Study of Seven Provinces in China

**DOI:** 10.3390/nu14153152

**Published:** 2022-07-30

**Authors:** Jieyu Liu, Di Gao, Yanhui Li, Manman Chen, Xinxin Wang, Qi Ma, Tao Ma, Li Chen, Ying Ma, Yi Zhang, Jun Ma, Yanhui Dong

**Affiliations:** 1Institute of Child and Adolescent Health, School of Public Health, Peking University, Beijing 100191, China; jieyulynne@163.com (J.L.); gaodi1993@163.com (D.G.); yanhui_lyh@163.com (Y.L.); 1911210173@pku.edu.cn (M.C.); 18702110295@163.com (Q.M.); 1610306216@pku.edu.cn (T.M.); clcl@bjmu.edu.cn (L.C.); mypku232@163.com (Y.M.); 1710306140@pku.edu.cn (Y.Z.); 2National Health Commission Key Laboratory of Reproductive Health, Beijing 100191, China; 3School of Public Health and Management, Ningxia Medical University, Yinchuan 750004, China; wangxinxin291314@163.com

**Keywords:** high blood pressure, breastfeeding duration, dietary behaviors, children and adolescents, China

## Abstract

This study was aimed to investigate the associations between breastfeeding duration and blood pressure (BP) levels, BP Z scores and high BP (HBP) in children and adolescents. A total of 57,201 participants including 29,491 boys and 27,710 girls aged 7–18 years were recruited from seven provinces in China in 2012. HBP was defined as BP levels of ≥95th percentiles of the referent age-, sex-, and height-specific population. Breastfeeding duration was divided into non-breastfeeding, 0–5 months, 6–12 months, and >12 months. Information on demographic, parental or family factors and dietary behaviors was collected through a self-administered questionnaire. Multivariable linear regression and logistic regression models were applied to assess the relationships of breastfeeding duration with BP levels and BP Z scores and with HBP, respectively. Stratified analyses were performed to further investigate the potential subgroup-specific associations. The reported prolonged breastfeeding (>12 months) rate was 22.53% in the total population. After full adjustment, compared to the non-breastfeeding group, breastfeeding for 6–12 months was correlated with 0.43 (95% CI: −0.75, −0.11) and 0.36 (95% CI: −0.61, −0.12) mmHg lower levels of SBP and DBP, respectively. Similar decrease trends were found for BP Z scores. Prolonged breastfeeding (>12 months) was associated with 1.33 (95% CI: 1.12, 1.58) and 1.12 (95% CI: 0.94, 1.33) higher odds of HBP in boys and girls, respectively. Based on nationally representative data, there was no evidence that a longer duration of breastfeeding is protective against childhood HBP. Breastfeeding for 6–12 months may be beneficial to BP, while prolonged breastfeeding durations might increase the odds of HBP in children and adolescents.

## 1. Introduction

Pediatric high blood pressure (HBP) represents a major public health threat. According to the latest findings, 22.1% of Chinese children and adolescents aged 13–17 years had HBP [1,2]. It is widely known that the persistent elevation of blood pressure (BP) is an established risk factor for target organ damage and cardiometabolic disorders [3,4]. The cause of HBP is likely multifactorial, related to socioeconomic and perinatal factors, familial aggregation, and unhealthy lifestyles. Since BP levels can be tracked from childhood to adulthood [5] and childhood HBP is an independent risk predictor of adult hypertension [6], timely identifying modifiable risk factors for HBP that can be addressed by early interventions is a top research priority.

Breastfeeding is an important means to provide optimal nutrients for infants’ healthy growth and development. Based on World Health Organization (WHO) recommendations, exclusive breastfeeding is suggested to be followed at least 6 months and continued until 2 years of age [7]. However, the overall breastfeeding rates remain low and the association of breastfeeding and BP in children and adolescents has yielded controversial conclusions. Some evidence has suggested that breastfeeding has a protective effect against HBP in children in Japan [8], Canada [9], United Kingdom [10], The Netherlands [11] and Brazil [12], and the protective effects have been shown to be independent of the duration of breastfeeding [11]. However, Nobre and his colleagues concluded that preschoolers breastfed for less than 6 months were more likely to develop HBP compared to those breastfed for a longer period [12]. On the contrary, an increased duration of breastfeeding was not associated with lowered adolescent HBP risk in a randomized intervention [13], which was in accordance with some systemic reviews [14,15]. In brief, a U-shaped association between the duration of breastfeeding and systolic BP was observed in data from five cohorts in Brazil, Guatemala, India, Philippines and South Africa [16]; the inconsistency between previous evidence on such relationships may be explained by different ethnicities, ages, definitions and groups of breastfeeding duration, and BP measurements.

There is a lack of evidence in developing settings such as China concerning breastfeeding and childhood BP. Compared to non-breastfed children, breastfeeding for ≥3 months was not associated with BP in a Hong Kong Chinese birth cohort [17], but it was demonstrated that breastfeeding for ≥6 months could partially decrease the risk of HBP in preschoolers with obesity status in Zhuhai city [18]. However, children aged 6~12 years with a breastfeeding duration of >10 months were shown to have a significantly increased prevalence of hypertension in Chongqing city [19]. In view of the current unsatisfactory situation that the peak HBP rates are trending increasingly towards younger ages in China and previous, inconsistent evidence mainly focused on single areas with different social infrastructures and basic postnatal characteristics, there is an urgent need for the effective assessment of nationally representative data to estimate the effects of different durations of breastfeeding on childhood BP.

To fill this gap, based on the representative data from seven provinces or cities in China, the main objective of the present study was to investigate the associations of breastfeeding duration with BP levels, BP Z scores and the odds of HBP in children and adolescents. Additionally, we tested the robustness of the results based on a series of stratified analyses.

## 2. Materials and Methods

### 2.1. Study Population

The data were obtained from a cross-sectional survey from seven provinces or cities in China (Hunan, Ningxia, Tianjin, Chongqing, Liaoning, Shanghai, and Guangzhou; registration number: NCT02343588). The full trial protocol (sampling procedure and measurements) was previously published [20]. Briefly, we selected the original population using a multi-stage cluster random sampling method. We randomly chose several regions from each province, and we chose approximately 12–16 schools randomly comprising primary schools, junior high schools, and middle high schools in each region. Then, we randomly selected two classes in each grade from each school. All the students and their parents from selected classes were invited. According to the inclusion and exclusion criteria, children and adolescents with missing information of breastfeeding duration, anthropometric measurements, and BP measurements were further excluded. Finally, a total of 57,201 children and adolescents aged 7–18 years (29,491 boys and 27,710 girls) with available data were included for the final analysis. The study was approved by the ethical committee of the Peking University (number: IRB0000105213034). All students participated, and their parents signed informed consent forms.

### 2.2. Anthropometric Measurement

Trained project members conducted anthropometric measurements based on a standardized procedure, and the measuring instruments were the same at all study sites. Height was measured twice using a portable stadiometer (model TZG, China) to the nearest 0.1 cm, with the students standing upright and barefoot. Weight was measured using a lever-type weight scale (model RGT-140, China) to the nearest 0.1 kg, with students only wearing light clothes and no shoes. We calculated the average level of the two measurements for final analyses. Body mass index (BMI) was calculated as body weight (kg) divided by height (m) squared. Overweight and obesity were classified using the sex- and age-specific BMI reference values developed by the International Obesity Task Force (IOTF) [21].

### 2.3. BP Measurement and Outcome Variables

BP was measured using an auscultation mercury sphygmomanometer (model XJ1ID, China). Three cuff sizes (7, 9, and 12 cm width) were selected according to the mid-upper arm circumference of the children, which stipulated that the cuff bladder width should cover 40% of the mid-arm circumference. In severely overweight adolescents, it would have been necessary to use cuffs larger than those available. The cuff was placed ~2 cm above the crease of the elbow. Students were asked to sit comfortably for at least 10 min prior to the first reading. Systolic blood pressure (SBP) was defined as the onset of “tapping” Korotkoff sound (K1), and diastolic blood pressure (DBP) was defined as the fifth Korotkoff sound (K5). BP was measured twice, with a 1 min gap between replicates. If the measured difference was >10 mmHg, measurement was repeated until the final two measures differed by ≤10 mmHg, and the average of SBP and DBP in the final two measures was used in analyses. The stadiometers, scales and auscultation mercury sphygmomanometer were calibrated, and the same instruments were used at all surveyed schools. We rechecked all anthropometric and BP measurements in 5% of subjects daily. If the proportion of invalid cases exceeded 10%, all the measures of that day were considered invalid and were measured again.

Age-, sex-, and height-specific BP Z scores were calculated according to the reference range of the National High Blood Pressure Education Program Working Group on High Blood Pressure in Children and Adolescents [22]. HBP was defined as SBP and/or DBP levels of children and adolescents ≥ the age-, sex-, and height-specific 95th percentile references [22]. The primary outcomes were BP levels, BP Z scores and HBP.

### 2.4. Assessment of Demographic Variables and Breastfeeding Duration

We collected the basic information and lifestyle behaviors from each child’s questionnaire. Additionally, a parental self-administered questionnaire was used to collect information about demographic, neonatal, parental or family characteristics. To obtain more accurate information, questionnaires of children grades 1–3 were filled in by parents. Children above the fourth grade filled in their questionnaire by themselves, as instructed by the class teacher. Before the survey, all eligible investigators were involved in a training session to become familiar with the whole process.

For demographic factors, parents were required to record their children’s birth weight based on the record given by the birth certificate or by health clinic. Single-child status was classified as “yes” or “no”. Residence area was grouped into “rural area” and “urban area”. Parents were also asked to provide information on feeding (breastfeeding or not) and its duration (in month), which was divided into: non-breastfeeding, 0–5 months, 6–12 months, and >12 months [23]. The non-breastfeeding type included cow’s milk, goat’s milk, and formulated milk. We defined breastfeeding >12 months as a prolonged breastfeeding duration. For parental or family characteristics, parents were asked to report the maternal age of delivery and family history of diseases including hypertension, diabetes, heart diseases, cerebrovascular diseases and obesity. Parental educational attainment was classified as “primary school or below”, “secondary or equivalent” and “junior college or above”. Monthly household income was defined as the sum of monthly income (in Chinese yuan, CNY) of all household members and then classified as <5000, ≥5000 CNY, or refuse to answer.

### 2.5. Measurement and Definition of Lifestyle Behaviors

As previously published [20,24,25,26], the frequency (days) and serving per day of dietary behaviors, including the consumption of fruits, vegetables, SSBs, and meat over the past 7 days, were surveyed. Participants were asked “How many days have you eaten fruits/vegetables/meat or drunk SSB over the past 7 days? How many servings in one day?”. To better understand the intake of fruits/vegetables, one serving was defined as the size of an ordinary adult’s closed fist and roughly equaled a medium-sized apple (≈200 g) [27], as described previously [24,25]. SSBs included Coca-Cola, Sprite, orange juice, Nutrition Express, and Red Bull [26]. One serving of SSB was classified as a canned beverage (approximately 250 mL), and one portion of meat equaled an adult’s palm (approximately 100 g) [28]. The daily dietary intake was calculated as: average daily intake  =  (frequencies of consumption × servings in those days)/7. According to the dietary guidelines for school-age children in China (2016), optimum dietary components were defined as fruits of ≥1.5 servings/day, vegetables of ≥2 servings/day, meat products of 2–3 servings/day, and SSB of <1 serving/week [29].

Information about each child’s physical activity was collected based on the International Physical Activity Questionnaire-Short Form (IPAQ-SF) [30]. All recruited students reported their frequency (days) and duration (hours and minutes) of moderate-to-vigorous-intensity physical activities (MVPA) in the past 7 days, and the average time was calculated as: average daily time = (days of MVPA × duration in those days)/7.

### 2.6. Statistical Analysis

Continuous variables were expressed as mean ± standard deviation (SD), and categorical variables were expressed as numbers and percentages (*n*, %). Demographic and other characteristics by different breastfeeding durations in each sex were examined with a one-way analysis of variance (ANOVA) test for continuous variables and Pearson’s chi-squared test for categorical variables. A multivariable linear regression model was applied to assess the relationships of breastfeeding duration with BP levels and BP Z scores in each sex; meanwhile, a binary logistic regression model was applied to calculate odds ratio (OR) and 95% confidence interval (95% CI) in order to analyze the associations between breastfeeding duration and the odds of HBP, respectively. A fully adjusted model controlled for age, sex, birth weight, single-child status, overweight/obesity status, residence area, maternal age at delivery, parental educational attainment, family history of diseases (hypertension, diabetes, heart disease, cerebrovascular disease and obesity), monthly household income, dietary behaviors (including fruit, vegetable, SSB and meat consumption), and physical activity. Furthermore, a series of stratified analyses were performed to test the robustness of the main results based on age classification (7–10 years old, 11–14 years old, and 15–18 years old), overweight/obesity status, birth weight, single-child status, residence area and parental educational attainment. All statistical analyses were performed using Statistical Analysis System (SAS) software (version 9.4, SAS Institute, Cary, NC, USA), and a two-sided *p* < 0.05 was considered statistically significant.

## 3. Results

### 3.1. Characteristics of Study Population

General characteristics of the 57,201 children and adolescents included in final analysis are presented in Table 1 by breastfeeding group. The mean ages were 10.81 (SD: 3.02) years, 11.81 (SD: 3.25) years, 10.88 (SD: 3.00) years and 11.22 (SD: 2.95) years in the non-breastfeeding, 0–5 month, 6–12 month, and >12 month groups, respectively. Differences in age, BMI, birth weight, single-child status, residence area, and parental educational attainment between the four breastfeeding groups were detected (all *p* < 0.001). Similar significant differences were also found regarding family history of diseases, monthly household income and eating behaviors in various breastfeeding groups (all *p* < 0.001).

### 3.2. Association between Breastfeeding Duration and Blood Pressure

Table 2 shows the differences in BP levels and BP Z scores among the four breastfeeding duration groups before and after full adjustment. The average SBP and DBP levels were, respectively, 106.66 ± 12.32 mmHg and 67.34 ± 8.85 mmHg in boys and 103.61 ± 11.45 mmHg and 66.19 ± 8.50 mmHg in girls. Furthermore, the mean BP levels remained the lowest in the 6–12-month breastfeeding group and obviously increased in the group of >12 months of breastfeeding. Regarding breastfeeding as a continuous variable, breastfeeding duration was positively correlated with higher levels of SBP and DBP in both sexes (*p* <0.05). Regarding categorical variables, when the non-breastfeeding group was regarded as the reference, a prolonged breastfeeding duration was associated with higher levels of SBP (β: 1.97, 95% CI: 1.64, 2.31) and DBP (β: 1.18, 95% CI: 0.93, 1.43) in the total population after full adjustment. These positive associations were more pronounced in girls (SBP: β: 2.20, 95% CI: 1.73, 2.67; DBP: β: 1.36, 95% CI: 1.00, 1.71). Surprisingly, compared to the non-breastfeeding group, breastfeeding for 6–12 months was correlated with 0.43 (95% CI: −0.75, −0.11) and 0.36 (95% CI: −0.61, −0.12) mmHg lower levels of SBP and DBP levels, respectively, which was more evident in boys. Similar inverse correlations were also found in girls, though they failed to reach significance. The findings for BP Z scores were similar to the BP levels in both boys and girls.

Consistent with the findings of linear regression, after controlling for potential confounders, the odds of HBP increased as the duration of breastfeeding increased (Table 3). Notably, both in boys and girls, a lower prevalence of HBP in the 6–12-month breastfeeding group (boys: 12.54%; girls: 12.34%) and a higher HBP prevalence in the group of breastfeeding for more than 12 months (boys: 19.87%; girls: 18.35%) were observed. A prolonged breastfeeding duration was significantly correlated with HBP (OR: 1.21, 95% CI: 1.08, 1.37), especially in boys (OR: 1.33, 95% CI: 1.12, 1.58).

### 3.3. Subgroup Analyses

A prolonged breastfeeding duration could also increase the odds of overweight/obesity in children and adolescents (Appendix A), so we also divided the population according to several potential covariates to test the robustness of the findings. Notably, children and adolescents breastfed for 6–12 months had lower BP and BP Z scores in each subgroup (Appendix A). Prolonged breastfeeding was positively correlated with higher SBP and DBP almost in all subgroups (Appendix A). In addition, the prevalence of HBP in each subgroup is presented in Appendix A. Notably, the results did not significantly change in each age group, and a prolonged breastfeeding duration was still associated with higher odds of HBP in all groups (Figure 1). For example, a prolonged breastfeeding duration was positively correlated with the odds of HBP, independent of their weight status, though the odds of HBP in the overweight/obesity participants were more pronounced (OR: 1.37, 95% CI: 1.12, 1.68). In addition, these associations could be more evident in those without siblings, those from urban areas, and those with higher parental educational backgrounds, but the prolonged breastfeeding duration in all subgroups showed an overall similar trend towards increased HBP risks.

## 4. Discussion

To our knowledge, based on this nationally representative cross-sectional study in China, prolonged breastfeeding for >12 months could increase the levels of BP and BP Z scores, as well as the odds of HBP, in both boys and girls. However, breastfeeding for 6–12 months seems to be a protective factor for high BP. Since breastfeeding is important for the health of children and adolescents, the better allocation of healthcare and resources to promote breastfeeding for an appropriate duration would benefit them by reducing BP levels and their odds of developing HBP. In addition, the studied subgroup characteristics could be used for targeting infants and young children to improve breastfeeding and complementary feeding practices in China.

The results of the present study suggest that breastfeeding might be beneficial to BP but the benefits might begin to wane at the age of 12 months. We speculate that an appropriate duration (6–12 months) of infant and young child feeding could be beneficial to the BP of children and adolescents. However, the evidence inferred from previous literature has resulted in inconsistent conclusions. It has been widely acknowledged that longer breastfeeding has a protective effect against elevated BP even in young children, which subtle but important [8,10]. A fully adjusted model demonstrated a 0.2 mm Hg reduction in SBP for each 3 months of breastfeeding [10]. However, the authors of that study did not find a significant association between the duration of breastfeeding and BP in a school-based surveillance [31]. Meanwhile, non-significant associations between breast milk n-3 long-chain polyunsaturated fatty acids and BP were mechanistically detected in one individual participant’s meta-analysis [32]. This inconsistency could be explained by different study populations, data-collection methods, or other post-natal factors, e.g., childhood growth and feeding type. The beneficial effects of breast milk on BP may be explained by a high concentration of long-chain polyunsaturated fatty acids in milk, as they are important structural components of the cell membranes [33]. Second, breastfeeding strongly affects the composition of the intestinal microbiota [34,35], and altered intestinal microbiota might contribute to later-life atherogenic processes [36]. Still, the exact mechanism linking breastfeeding to BP is unknown, and it could be the long-term effects of lower sodium intake in infancy or of greater control over food intake and self-regulation in infancy, resulting in less vulnerability to HBP.

Overall, there is no persuasive evidence that longer durations of breastfeeding protect against childhood HBP. Notably, the previously reported U-shaped relationship between the duration of breastfeeding and HBP is some strong evidence [16] that partly agreed with our results. For the adverse effects of prolonged breastfeeding for over 12 months, the mechanism needs to be further investigated in future studies. The authors of one study concluded that fat increased and carbohydrates decreased significantly in infants who were breastfed for longer than 18 months [37]. In the Hertfordshire cohort, prolonged breastfeeding was also associated with high concentrations of serum cholesterol in later life [38]. Increased concentrations of fat and serum cholesterol are crucial predictors for elevated BP. Though we could not investigate the potential mechanisms in our observational study, the present epidemiological results can inform future research on this topic. Based on nationally representative data from a large sample size, we propose that appropriate lengths of time for breastfeeding should be encouraged instead of long-term breastfeeding in order to promote healthy BP levels in pediatric populations.

In addition, recommended practices from the World Health Organization (WHO) and United Nations International Children’s Emergency Fund (UNICEF) include the timely introduction of complementary foods at 6 months of age, sufficient meal frequency and portions sizes, and diversity of diet [39]. Additionally, the authors of one study in China suggested that complementary foods should be introduced at around 6 months of age when considering infant developmental readiness [40]. Generally, after the addition of complementary food, the progression from liquid food to family foods occurs in a period from 6 to 24 months of age. During this period, children should consume foods and beverages from at least five out of eight defined food groups each day. Therefore, the type and duration of postnatal feeding may also be a driver of infant growth and a modifiable determinant of cardiovascular health. In the current Chinese context in which the breastfeeding rate is low and complementary feeding practices are not optimal [41], breastfeeding for approximately 6 months is a desirable goal and the complementary foods should not be delayed beyond 6 months.

Since we assessed BP in children and adolescents before the onset of many traditional cardiovascular disease risk factors (e.g., smoking and alcohol consumption), some potential indicators seemed to be important factors affecting the occurrence of childhood HBP. In our study, the effects of prolonged breastfeeding durations were more evident in boys, which might validate previous findings. On the one hand, significant associations were detected between high n-3 LC-PUFA content and low BP in boys at 4 months but not in girls [42]. This might be explained by the fact that the physiologic sex-specific requirements of some hormones may regulate the leptin concentrations in breast milk differently [43]; however, how sex-specific responses to nutritional stimuli could contribute to BP fluctuation remains unclear. On the other hand, in males, but not females, age-adjusted SBP and DBP were significantly correlated with dietary sodium intake, indicating that male counterparts seemed to be more susceptible to dietary changes of complementary foods and later family foods after 12 months [44]. Apart from the sex differences, over-nutrition brought by single-child status (the focus of one family), urban residence, and higher parental educational background might contribute to the associations between breastfeeding and HBP. Future longitudinal prospective studies are needed to confirm the subgroup-related associations.

The obvious strengths of the current study was the nationally representative sample from seven provinces or cities in China, from which we focused on 7–18-year-old children and adolescents. However, several potential limitations should also be considered. Firstly, the prevalence of HBP may have been overestimated since the BP measurements were collected on one occasion [45]. In future studies, BP measurements taken over multiple visits or using ambulatory BP monitor would be better. Secondly, the information on breastfeeding duration and lifestyle behaviors was self-reported, which could have allowed for a certain degree of recall bias. However, we carried out strict quality control to ensure the reliability of this process, and the questionnaires were rechecked by 3% within one week; therefore, the quality of the information on breastfeeding duration was reliable. In addition, measuring breastfeeding is complex and involves feeding type, quantity, and the consumption of complementary foods. Since it was difficult to accurately collect detailed information on breastfeeding in a large population survey, we could not identify the specific type of feeding. Thirdly, some factors that influence the decision to breastfeed were not considered in our study. For example, maternal breastfeeding intentions before delivery or reasons for not breastfeeding, including insufficient economic support or undesirable breastfeeding experiences were not surveyed, but we deem it unlikely that these indicators could have affected such observed associations. Fourthly, although we adjusted for a few potential confounders, other important dietary factors influencing BP, such as salt and macronutrient intake, were not available, though residual confounding was still possible. Fifthly, we could not generate causal relationships of breastfeeding and BP due to the cross-sectional nature of the study. Future randomized controlled trials are necessary to confirm and determine the clinical implications of the present findings.

## 5. Conclusions

In conclusion, based on nationally representative data, the benefits of breastfeeding might start at the age of 6 months and begin to wane at 12 months. Since exclusive breastfeeding is suggested to be followed for at least six months by WHO recommendations, and we were not able to reach a consensus regarding the effects of longer durations of breastfeeding on children, we came to conclusions that the better allocation of healthcare and resources to promote breastfeeding for an appropriate duration instead of longer durations would benefit children by reducing BP levels and their odds of developing HBP.

## Figures and Tables

**Figure 1 nutrients-14-03152-f001:**
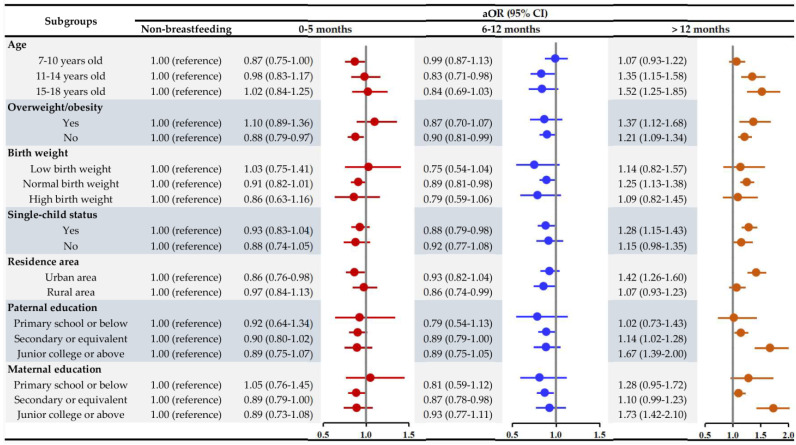
Associations between breastfeeding duration and high blood pressure in various subgroups. (Model: adjusted for age, sex, birth weight, single-child status, overweight/obesity status, residence area, maternal age at delivery, parental educational attainment, family history of diseases (hypertension, diabetes, heart disease, cerebrovascular disease and obesity), monthly household income, dietary behaviors (including fruit, vegetable, SSB and meat consumption), and physical activity).

**Table 1 nutrients-14-03152-t001:** Baseline characteristic of included sample divided by breastfeeding duration in boys and girls.

	Total Population	Non-Breastfeeding	0–5 Months	6–12 Months	>12 Months	*p*-Value
Population	57,201	7086 (12.39%)	21,953 (38.38%)	15,273 (26.70%)	12,889 (22.53%)	
Baseline demographic factors						
Age, year	11.30 ± 3.12	10.81 ± 3.02	11.81 ± 3.25	10.88 ± 3.00	11.22 ± 2.95	<0.001
Boys, n (%)	29,491 (51.56%)	3652 (51.54%)	11,735 (53.46%)	7585 (49.66%)	6519 (50.58%)	<0.001
BMI, kg/m^2^	18.79 ± 3.76	18.66 ± 3.86	18.87 ± 3.69	18.49 ± 3.59	19.08 ± 3.98	<0.001
Overweight/obesity, n (%)	5766 (10.08%)	784 (11.06%)	1998 (9.10%)	1369 (8.96%)	1615 (12.53%)	<0.001
Birth weight, g	3309.91 ± 509.47	3258.62 ± 532.07	3261.23 ± 522.58	3316.91 ± 493.18	3378.73 ± 493.64	<0.001
Birth weight group, n (%)						<0.001
Low birth weight	12,672 (22.15%)	724 (10.22%)	9956 (45.35%)	1177 (7.71%)	815 (6.32%)	
Normal birth weight	39,961 (69.86%)	5772 (81.46%)	10,874 (49.53%)	12,631 (82.70%)	10,684 (82.89%)	
High birth weight	4568 (7.99%)	590 (8.33%)	1123 (5.12%)	1465 (9.59%)	1390 (10.78%)	
Single-child, n (%)	40,633 (71.04%)	5085 (71.76%)	17,758 (80.89%)	10,285 (67.34%)	7505 (58.23%)	<0.001
Urban area, n (%)	37,739 (65.98%)	4950 (69.86%)	16,226 (73.91%)	10,078 (65.99%)	6485 (50.31%)	<0.001
Parental or family factors						
Maternal age at delivery, year	27.47 ± 71.14	27.36 ± 5.36	27.49 ± 67.01	27.01 ± 54.82	28.06 ± 103.01	0.702
Paternal educational attainment, n (%)						<0.001
Primary school or below	3539 (6.19%)	399 (5.63%)	913 (4.16%)	912 (5.97%)	1315 (10.20%)	
Secondary or equivalent	40,716 (71.18%)	4331 (61.12%)	17,281 (78.72%)	9694 (63.47%)	9410 (73.01%)	
Junior college or above	12,946 (22.63%)	2356 (33.25%)	3759 (17.12%)	4667 (30.56%)	2164 (16.79%)	
Maternal educational attainment, n (%)						<0.001
Primary school or below	4829 (8.44%)	558 (7.87%)	1247 (5.68%)	1312 (8.59%)	1712 (13.28%)	
Secondary or equivalent	40,579 (70.94%)	4323 (61.01%)	17,251 (78.58%)	9721 (63.65%)	9284 (72.03%)	
Junior college or above	11,793 (20.62%)	2205 (31.12%)	3455 (15.74%)	4240 (27.76%)	1893 (14.69%)	
Family history of diseases, n (%)						
Hypertension	3234 (5.65%)	531 (7.49%)	886 (4.04%)	977 (6.40%)	840 (6.52%)	<0.001
Diabetes	1092 (1.91%)	211 (2.98%)	305 (1.39%)	299 (1.96%)	277 (2.15%)	<0.001
Heart diseases	1097 (1.92%)	183 (2.58%)	306 (1.39%)	272 (1.78%)	336 (2.61%)	<0.001
Cerebrovascular diseases	513 (0.90%)	77 (1.09%)	133 (0.61%)	156 (1.02%)	147 (1.14%)	<0.001
Obesity	4248 (7.43%)	722 (10.19%)	1009 (4.60%)	1371 (8.98%)	1146 (8.89%)	<0.001
Monthly household income, n (%)						<0.001
<5000 CNY	16,535 (28.91%)	2160 (30.48%)	3940 (17.95%)	4586 (30.03%)	5849 (45.38%)	
≥5000 CNY	8022 (14.02%)	1211 (17.09%)	2047 (9.32%)	2712 (17.76%)	2052 (15.92%)	
Refuse to answer	32,644 (57.07%)	3715 (52.43%)	15,966 (72.73%)	7975 (52.22%)	4988 (38.70%)	
Dietary behaviors						
Fruits of ≥1.5 servings/day, n (%)	15,087 (26.38%)	2109 (29.76%)	4776 (21.76%)	4248 (27.81%)	3954 (30.68%)	<0.001
Vegetables of ≥2 servings/day, n (%)	23,682 (41.40%)	3196 (45.10%)	7637 (34.79%)	6814 (44.61%)	6035 (46.82%)	<0.001
meat products of 2–3 servings/day, n (%)	8078 (14.12%)	1153 (16.27%)	2776 (12.65%)	2711 (17.75%)	1438 (11.16%)	<0.001
SSB of <1 serving/week	34,298 (59.96%)	4033 (56.92%)	13,995 (63.75%)	8888 (58.19%)	7382 (57.27%)	<0.001
Physical activity, hour/day	0.40 ± 0.76	0.38 ± 0.73	0.42 ± 0.82	0.37 ± 0.69	0.42 ± 0.76	<0.001

Abbreviation: BMI, body mass index.

**Table 2 nutrients-14-03152-t002:** Multiple linear regression analysis of breastfeeding duration and BP and BP Z scores, β (95% CI).

Blood Pressure	Breastfeeding Duration	Categorical Variables			
		Non-Breastfeeding	0–5 Months	6–12 Months	>12 Months
Total population					
SBP (mmHg)	105.18 ± 12.01	104.07 ± 12.02	105.09 ± 12.07	103.66 ± 11.72	107.74 ± 11.81
*Unadjusted*	**0.42 (0.40, 0.45)**	1 (Reference)	**1.02 (0.71, 1.34)**	**−0.40 (−0.74, −0.07)**	**3.67 (3.32, 4.01)**
*Fully adjusted*	**0.28 (0.26, 0.31)**	1 (Reference)	−0.35 (−0.97, 0.28)	**−0.43 (−0.75, −0.11)**	**1.97 (1.64, 2.31)**
SBP Z-score	0.00 ± 1.00	−0.09 ± 1.00	−0.01 ± 1.01	−0.13 ± 0.98	0.21 ± 0.98
*Unadjusted*	**0.04 (0.03, 0.04)**	1 (Reference)	**0.09 (0.06, 0.11)**	**−0.03 (−0.06, −0.01)**	**0.31 (0.28, 0.33)**
*Fully adjusted*	**0.02 (0.02, 0.03)**	1 (Reference)	−0.04 (−0.06, 0.01)	**−0.04 (−0.06, −0.01)**	**0.16 (0.14, 0.19)**
DBP (mmHg)	66.78 ± 8.70	66.02 ± 8.47	66.91 ± 8.90	65.70 ± 8.57	68.27 ± 8.41
*Unadjusted*	**0.26 (0.24, 0.27)**	1 (Reference)	**0.89 (0.65, 1.12)**	**−0.32 (−0.57, −0.08)**	**2.25 (2.00, 2.50)**
*Fully adjusted*	**0.17 (0.15, 0.19)**	1 (Reference)	**−0.38 (−0.64, −0.13)**	**−0.36 (−0.61, −0.12)**	**1.18 (0.93, 1.43)**
DBP Z-score	0.00 ± 1.00	−0.09 ± 0.97	0.01 ± 1.02	−0.12 ± 0.98	0.17 ± 0.97
*Unadjusted*	**0.03 (0.03, 0.03)**	1 (Reference)	**0.10 (0.08, 0.13)**	**−0.04 (−0.07, −0.01)**	**0.26 (0.23, 0.29)**
*Fully adjusted*	**0.02 (0.02, 0.02)**	1 (Reference)	**−0.04 (−0.07, −0.01)**	**−0.04 (−0.07, −0.01)**	**0.14 (0.11, 0.16)**
Boys					
SBP (mmHg)	106.66 ± 12.32	105.61 ± 12.27	106.73 ± 12.35	105.12 ± 12.11	108.89 ± 12.21
*Unadjusted*	**0.39 (0.35, 0.42)**	1 (Reference)	**1.12 (0.66, 1.57)**	**−0.49 (−0.97, 0.00)**	**3.28 (2.78, 3.78)**
*Fully adjusted*	**0.27 (0.23, 0.30)**	1 (Reference)	−0.62 (−0.96, 0.03)	**−0.56 (−1.02, −0.11)**	**1.82 (1.35, 2.29)**
SBP Z-score	0.12 ± 1.03	0.04 ± 1.02	0.13 ± 1.03	0.00 ± 1.01	0.31 ± 1.02
*Unadjusted*	**0.03 (0.03, 0.04)**	1 (Reference)	**0.09 (0.06, 0.13)**	**−0.04 (−0.08, 0.00)**	**0.27 (0.23, 0.31)**
*Fully adjusted*	**0.02 (0.02, 0.03)**	1 (Reference)	−0.05 (−0.09, 0.01)	**−0.05 (−0.08, −0.01)**	**0.15 (0.11, 0.19)**
DBP (mmHg)	67.34 ± 8.85	66.63 ± 8.57	67.57 ± 9.05	66.21 ± 8.71	68.64 ± 8.61
*Unadjusted*	**0.23 (0.21, 0.26)**	1 (Reference)	**0.94 (0.62, 1.27)**	**−0.42 (−0.76, −0.07)**	**2.01 (1.66, 2.37)**
*Fully adjusted*	**0.16 (0.13, 0.18)**	1 (Reference)	**−0.49 (−0.85, −0.13)**	**−0.54 (−0.88, −0.20)**	**1.03 (0.67, 1.39)**
DBP Z-score	0.06 ± 1.02	−0.02 ± 0.98	0.09 ± 1.04	−0.07 ± 1.00	0.21 ± 0.99
*Unadjusted*	**0.03 (0.02, 0.03)**	1 (Reference)	**0.11 (0.07, 0.15)**	**−0.05 (−0.09, −0.01)**	**0.23 (0.19, 0.27)**
*Fully adjusted*	**0.02 (0.02, 0.02)**	1 (Reference)	**−0.06 (−0.10, −0.02)**	**−0.06 (−0.10, −0.02)**	**0.12 (0.08, 0.16)**
Girls					
SBP (mmHg)	103.61 ± 11.45	102.42 ± 11.52	103.21 ± 11.46	102.22 ± 11.13	106.55 ± 11.26
*Unadjusted*	**0.46 (0.42, 0.49)**	1 (Reference)	**0.79 (0.35, 1.22)**	−0.20 (−0.66, 0.25)	**4.13 (3.66, 4.60)**
*Fully adjusted*	**0.29 (0.26, 0.33)**	1 (Reference)	−0.17 (−0.65, 0.30)	−0.24 (−0.69, 0.21)	**2.20 (1.73, 2.67)**
SBP Z-score	−0.13 ± 0.95	−0.23 ± 0.96	−0.16 ± 0.95	−0.25 ± 0.93	0.11 ± 0.94
*Unadjusted*	**0.04 (0.04, 0.04)**	1 (Reference)	**0.07 (0.03, 0.10)**	−0.02 (−0.05, 0.02)	**0.34 (0.30, 0.38)**
*Fully adjusted*	**0.02 (0.02, 0.03)**	1 (Reference)	−0.01 (−0.05, 0.03)	−0.02 (−0.06, 0.02)	**0.18 (0.14, 0.22)**
DBP (mmHg)	66.19 ± 8.50	65.38 ± 8.31	66.15 ± 8.67	65.19 ± 8.40	67.89 ± 8.19
*Unadjusted*	**0.28 (0.25, 0.30)**	1 (Reference)	**0.77 (0.44, 1.10)**	−0.19 (−0.53, 0.15)	**2.51 (2.16, 2.86)**
*Fully adjusted*	**0.17 (0.15, 0.20)**	1 (Reference)	−0.25 (−0.61, 0.12)	−0.17 (−0.51, 0.17)	**1.36 (1.00, 1.71)**
DBP Z-score	−0.07 ± 0.98	−0.16 ± 0.95	−0.07 ± 1.00	−0.18 ± 0.97	0.13 ± 0.94
*Unadjusted*	**0.03 (0.03, 0.03)**	1 (Reference)	**0.09 (0.05, 0.13)**	−0.02 (−0.06, 0.02)	**0.29 (0.25, 0.33)**
*Fully adjusted*	**0.02 (0.02, 0.02)**	1 (Reference)	−0.03 (−0.07, 0.01)	−0.02 (−0.06, 0.02)	**0.16 (0.11, 0.20)**

Fully adjusted model: adjusted for age, sex, birth weight, single-child status, overweight/obesity status, residence area, maternal age at delivery, parental educational attainment, family history of diseases (hypertension, diabetes, heart disease, cerebrovascular disease and obesity), monthly household income, dietary behaviors (including fruit, vegetable, SSB and meat consumption), and physical activity. SBP: systolic blood pressure; DBP: diastolic blood pressure. Bold values refer to *p* < 0.05.

**Table 3 nutrients-14-03152-t003:** Multivariate odds ratios (ORs) and 95% confidence intervals (CIs) of HBP by breastfeeding duration.

Breastfeeding Duration	Population (%)	HBP Prevalence (%)	ORs (95% CI)	
			Unadjusted	Fully Adjusted
**Total population**	57,201	8709 (15.23)		
Non-breastfeeding	7086 (12.39)	986 (13.91)	1 (Reference)	1 (Reference)
0–5 months	21,953 (38.38)	3359 (15.30)	**1.12 (1.04–1.21)**	0.94 (0.82–1.07)
6–12 months	15,273 (26.70)	1900 (12.44)	**0.88 (0.81–0.96)**	**0.87 (0.76–0.99)**
>12 months	12,889 (22.53)	2464 (19.12)	**1.46 (1.35–1.59)**	**1.21 (1.08–1.37)**
**Boys**	29,491	4660 (15.80)		
Non-breastfeeding	3652 (12.38)	525 (14.38)	1 (Reference)	1 (Reference)
0–5 months	11,735 (39.79)	1889 (16.10)	**1.14 (1.03–1.27)**	1.01 (0.84–1.21)
6–12 months	7585 (25.82)	951 (12.54)	**0.85 (0.76–0.96)**	0.84 (0.70–1.01)
>12 months	6519 (22.11)	1295 (19.87)	**1.48 (1.32–1.65)**	**1.33 (1.12–1.58)**
**Girls**	27,710	4049 (14.61)		
Non-breastfeeding	3434 (12.39)	461 (13.42)	1 (Reference)	1 (Reference)
0–5 months	10,218 (36.87)	1470 (14.39)	1.08 (0.97–1.21)	0.88 (0.73–1.06)
6–12 months	7688 (27.74)	949 (12.34)	0.91 (0.81–1.02)	0.90 (0.75–1.08)
>12 months	6370 (22.99)	1169 (18.35)	**1.45 (1.29–1.63)**	1.12 (0.94–1.33)

Fully adjusted model: adjusted for age, sex, birth weight, single-child status, overweight/obesity status, residence area, maternal age at delivery, parental educational attainment, family history of diseases (hypertension, diabetes, heart disease, cerebrovascular disease and obesity), monthly household income, dietary behaviors (including fruit, vegetable, SSB and meat consumption), and physical activity. HBP: high blood pressure. Bold values refer to *p* < 0.05.

## Data Availability

The raw data supporting the conclusions of this article will be made available by the authors without undue reservation.

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
