# Peer review of "Breastfeeding Duration and High Blood Pressure in Children and Adolescents: Results from a Cross-Sectional Study of Seven Provinces in China"

_nutrients, 2022, doi:10.3390/nu14153152_

Round 1
Reviewer 1 Report
Breastfeeding protects infants from infections. Long-term effects on cardiovascular health are less clar. From a life-course perspective, the predominant paradigm has considered fetal and infant growth as key critical periods that program metabolism for life. The type and quantity of postnatal feeding may be a driver of infant growth and a modifiable determinant of cardiovascular health.
Elevated blood pressure level during childhood and adolescence is a recognized predictor of adulthood blood pressure, leading to increased CVD disease in adults. It is imperative to study the prevalence and risk factors for childhood adolescents´hypertension.
The exact mechanism linking breastfeeding to blood pressure is unknown, but it could be the long-term effects of lower sodium intake in infancy or of greater control over food intake and self-regulation in infancy, resulting in less vulnerability of obesity.
Meta-analyses of observational studies suggest that breastfeeding has a modest effect on subsequent blood pressure, of about 1.1-1.4 mm Hg lower systolic blood pressure and 0.4-0.5 mm Hg lower diastolic blood pressure, for those breastfed compared with formula fed.
This study aimed to examine the associations between breastfeeding duration and blood pressure and to further investigate whether advocating healthy dietary behaviors could modify the potential adverse effects of prolonged breastfeeding. A U-shape relationship was established between breastfeeding duration and childhood. Breastfeeding for 6-12 months may be beneficial to blood pressure, while prolonged breastfeeding duration might increase the risk of high blood pressure in children and adolescents.
Reccomendations
L 178 When using the abbreviation CNY for the first time, please add the explanation Chinese yuan renmimbi between parantheses.
L 197 In table 1, under dietary behaviors, you have included physical activity (hour/day). Please explain how this was evaluated.
L 303 Please present a figure of the U-shape relationship between breastfeeding duration and childhood BP levels, Bp Z scores and HBP.
L395 The study has included a large segment of children and youth (7 -18 year-old children and adolescents). It could be said that during this period, people experience major fluctuations in dietary behaviors. Diet estimation (data recall of 7 days) could be a potential pathway for errors.
Author Response
“Breastfeeding Duration and High Blood Pressure in Children and Adolescents: Results from a Cross-Sectional Study of Seven Provinces in China”
Manuscript Number: 1817011
Rebuttal letter
Dear Dr. Lona Li:
Thank you very much for your letter and constructive comments. We have revised the manuscript and would like to re-submit for your consideration. The amendments are highlighted in red in the revised manuscript, and the point by point responses to the reviewer’s comments are listed below.
#Reviewer 1:
Breastfeeding protects infants from infections. Long-term effects on cardiovascular health are less clar. From a life-course perspective, the predominant paradigm has considered fetal and infant growth as key critical periods that program metabolism for life. The type and quantity of postnatal feeding may be a driver of infant growth and a modifiable determinant of cardiovascular health.
Elevated blood pressure level during childhood and adolescence is a recognized predictor of adulthood blood pressure, leading to increased CVD disease in adults. It is imperative to study the prevalence and risk factors for childhood adolescents´hypertension.
The exact mechanism linking breastfeeding to blood pressure is unknown, but it could be the long-term effects of lower sodium intake in infancy or of greater control over food intake and self-regulation in infancy, resulting in less vulnerability of obesity.
Meta-analyses of observational studies suggest that breastfeeding has a modest effect on subsequent blood pressure, of about 1.1-1.4 mm Hg lower systolic blood pressure and 0.4-0.5 mm Hg lower diastolic blood pressure, for those breastfed compared with formula fed.
This study aimed to examine the associations between breastfeeding duration and blood pressure and to further investigate whether advocating healthy dietary behaviors could modify the potential adverse effects of prolonged breastfeeding. A U-shape relationship was established between breastfeeding duration and childhood. Breastfeeding for 6-12 months may be beneficial to blood pressure, while prolonged breastfeeding duration might increase the risk of high blood pressure in children and adolescents.
Response: Thanks for your valuable suggestion and comments.
First, we agreed with the suggestion that the type and quantity of postnatal feeding may be a driver of infant growth and a modifiable determinant of cardiovascular health, regrettably, due to the limited data we could not analyze the type and quantity of breastfeeding. In addition, we believe that lower sodium intake in infancy or of greater control over food intake and self-regulation in infancy might result in less vulnerability of obesity, we added these points in Discussion as follows:
Discussion: Still, the exact mechanism linking breastfeeding to BP is unknown, but it might also be the long-term effects of lower sodium intake in infancy or of greater control over food intake and self-regulation in infancy, resulting in less vulnerability of HBP. (line 285-288)
Also, one study in China suggested that complementary foods should be introduced at around 6 months of age, taking infant developmental readiness into account[40]. Generally, After the addition of the complementary food, the progression from liquid food to family foods occurs in the period from 6 to 24 months of age. During this period, children should consume foods and beverages from at least five out of eight defined food groups each day. Therefore, the type and duration of postnatal feeding may also be a driver of infant growth and a modifiable determinant of cardiovascular health. (line 307-314)
Limitations: In addition, measuring breastfeeding was complex, involving feeding type, quantity, as well as the consumption of complementary foods. Since it was difficult to collect detailed information accurately on breastfeeding in a large population survey, we could not identify the specific type of feeding. (line 345-349)
Fourthly, although we adjusted for a few potential confounders, other important dietary factors influencing BP, such as salt and macronutrient intakes, were not available, residual confounding was still possible. (line 353-355)
Recommendations
L 178 When using the abbreviation CNY for the first time, please add the explanation Chinese yuan renmimbi between parantheses.
Response: Thanks for your suggestion. We have added the explanation Chinese yuan for CNY. The amendment is as follows:
Monthly household income was defined as the sum of monthly income (in Chinese yuan, CNY) of all household members and then classified into <5000, ≥5000 CNY, or refuse to answer. (line 152-154)
L 197 In table 1, under dietary behaviors, you have included physical activity (hour/day). Please explain how this was evaluated.
Response: Thanks for your comments. We have added the description of the information collecting about physical activity in the Methods:
Information about the child’s physical activity was collected based on the International Physical Activity Questionnaire-Short Form (IPAQ-SF) [30]. All recruited students reported their frequency (days) and duration (hours and minutes) of moderate to vigorous-intensity physical activities (MVPA) in the past 7 days, and the average time was calculated as: average daily time = (days of MVPA × duration in those days)/7. (line 170-174)
L 303 Please present a figure of the U-shape relationship between breastfeeding duration and childhood BP levels, Bp Z scores and HBP.
Response: Thanks for your comments. We rethink the results carefully, and come to the conclusion that a U-shaped curve is not much accurate for the association between breastfeeding duration and childhood BP. In a very large sample, after correcting for numerous confounding factors and having non-breastfed as a reference, those breastfed for more than one year have worse BP values, while those in the other two groups (breastfed only up to 5 months and breastfed for 6-12 months) breast milk would have almost no effect on BP. Relative categorical analysis on subjects with BP values above the 95th percentile (Table 3) also demonstrates a statistically significant correlation only in those breastfed for more than one year, while only a non-significant trend, positive for males and negative for females, in the other two groups of breastfed. Therefore, given the uncertainty of the data of the eventual descending part of the curve, we revised throughout the manuscript and came to a conclusion as follows:
Based on national representative data, there was no evidence that longer duration of breastfeeding is protective against childhood HBP. Breastfeeding for 6-12 months may be beneficial to BP, while prolonged breastfeeding duration might increase the odds of HBP in children and adolescents. (line 31-34)
L395 The study has included a large segment of children and youth (7 -18 year-old children and adolescents). It could be said that during this period, people experience major fluctuations in dietary behaviors. Diet estimation (data recall of 7 days) could be a potential pathway for errors.
Response: Thanks for your valuable comment.
We agreed that people experience major fluctuations in dietary behaviors during 7-18 year-old, and we mentioned in the limitation that the dietary data recall of 7 days might not represent long-term dietary behaviors and may bring a certain degree of recall bias. The revision was as follows:
Secondly, the information of breastfeeding duration and lifestyle behaviors was self-reported, which could bring a certain degree of recall bias. However, we carried out strict quality control to ensure the reliability in this process, and the questionnaires would be rechecked by 3% within one week, the quality of the information on breastfeeding duration therefore could be reliable. (line 341-345)
After carefully consideration, we decided to delete the part of the modified effects of dietary behaviors in the manuscript, and focused on the topic of breastfeeding and childhood BP, BP Z scores and odds of HBP both in boys and girls, and we added stratified analyses to test the robustness of the results. The revised title is as follows:
“Breastfeeding Duration and High Blood Pressure in Children and Adolescents: Results from a Cross-Sectional Study of Seven Provinces in China”
To date, the evidence in developing settings such as China is still scarce. In view of the current unsatisfactory situation that the peak of HBP rates is trending increasingly towards younger ages in China, and the previous limited but inconsistent evidence mainly focused on a single area with different standard of living, social infrastructure, and postnatal characteristics, there is an urgent need of effective assessment of national representative data to estimate the effects of duration of breastfeeding on BP. So our research is still valuable for this field.
To fill this gap, based on the representative data from seven provinces or cities in China, the main objective of the present study was to investigate the associations between breastfeeding duration with BP levels, BP Z scores and the odds of HBP in children and adolescents.
Thanks for your valuable suggestion, based on your comments, the paper is now much improved!

Reviewer 2 Report
The manuscript "Dietary Behaviors May Modify the Association between Breastfeeding Duration and High Blood Pressure in Children and Adolescents: A National Cross-Sectional Study in China" aims to correlate blood pressure values at age 7-18 years with breastfeeding duration and other aspects of feeding behavior in a large population of nearly 30,00 children and adolescents. The results are intriguing because the authors come to the conclusion that prolonging breastfeeding beyond 12 months would have a negative effect on blood pressure, calling into question, at least in this respect, the widely held view that breast milk constitutes some sort of magic potion for the prevention of almost all diseases, including hypertension. The authors themselves do not escape this positive preconception as they state in their conclusions that: "the benefits of breastfeeding might start at the age of 6 months and begin to wane around the age of 12 months" while in the results they show that, compared to non-breastfed, the group of breastfed for 6-12 months have only a modest statistically significant positive effect only on diastolic blood pressure and only in males, while in the group of breastfed only up to 5 months the only significant finding is a small worsening in females. To summarize, in a very large sample, after correcting for numerous confounding factors and having non-breastfed as a reference, those breastfed for more than one year have worse blood pressure values, while those in the other two groups (breastfed only up to 5 months and breastfed for 6-12 months) breast milk would have almost no effect on blood pressure. Relative categorical analysis on subjects with blood pressure values above the 90th percentile (Table 3) also demonstrates a statistically significant correlation (in a worsening sense, i.e., an increase in OR) only in those breastfed for more than one year, while only a nonsignificant trend, positive for males and negative for females, in the other two groups of breastfed. Therefore, I do not think it is correct to speak of a U-shaped curve given the uncertainty of the data of the eventual descending part of the curve. The study has a few problems. In particular due to the fact that the percentage of subjects with high blood pressure values (above the 90th percentile) is very high. In theory, only 10% of children with values above the 90th percentile should be expected but, for many reasons, even a relatively large deviation from these values might be acceptable. In the manuscript the deviation appears excessive being at least double, and in some cases, almost triple the expected (Table 3). This may have some explanations that should be discussed and possibly put into the limitations of the study. The AAP guideline 2017 Clinical Practice Guideline for Screening and Management of High Blood Pressure in Children and Adolescents Flynn J. Volume 140, Issue 3 September 2017 gives the following guidance: “if the initial BP is elevated (≥90th percentile), providers should perform 2 additional blood pressure measurements at the same visit and average them. If using auscultation, this averaged measurement is used to determine the child's blood pressure category (normal, elevated blood pressure, stage 1 hypertension, or stage 2 hypertension)”. Thus, having used the average of only two first measurements in each child, it is not possible to confirm the diagnosis of elevated blood pression (or hypertension if the values were above the 95th percentile). Furthermore, it is known that subsequent measurements tend to give lower values than the first ones. Despite this, the prevalence of children/adolescents with elevated values in the sample seems excessive and should be commented on. Doubts may also arise as to how blood pressure is taken. With respect to the choice of cuff to be used in each child, "The fourth report on the diagnosis, evaluation, and treatment of high blood pressure in children and adolescents. Pediatrics. 2004, 114, 555-576" recommends, "by convention, an appropriate cuff size is a cuff with an inflatable bladder width that is at least 40 percent of the arm circumference at a point midway between the olecranon and the acromion," whereas the manuscript methods mention 50-75 percent of the arm circumference (line 137) which would suggest the use of cuffs that are too large in many cases, but this would have resulted in an underestimation of the actual blood pressure values. Conversely, in severely overweight adolescents it would have been necessary to use cuffs larger than those available and, in this case, the values would be overestimated. Finally, age is not a criterion for selecting the headset (line136). Also, for proper interpretation of pressure values, it would be more correct to parameterize the stature percentile to US standards and not to Chinese height standards (line 153), since US pressure percentiles are used. But these latter are marginal observations. Despite these observations about the suboptimal accuracy of the measurements, as pressure was measured the same way for everyone, the interesting finding of the association between prolonged breastfeeding and increased blood pressure values at 7-18 years remains. The part of the manuscript concerning the effects on blood pressure of more or less healthy dietary behaviors also seems to me unsatisfactory in both writing and content. In lines 31-32 of the abstract the statement that "the risk for high blood pressure brought by prolonged breastfeeding decreased in boys (OR=1.29, 95% CI: 0.83, 2.00)" is incomprehensible without reference to the baseline value of non-breastfed children. The same goes for line 285 where "more pronuonced" should have a comparison term. The comments to Figure 2 (which should be called a table) for the part about boys are acceptable. However, it is puzzling that there is no difference between boys having 1,2 or > 3 healty dietary behaviors. While in females the results have such a random pattern that it suggests that the data were not properly collected. In fact, it is a paradox that girls who have more healty dietary behaviors would have a higher risk of having high blood pressure values. The results of this part of the study are so uncertain that it would be preferable not to cite them in the abstract, or even delete them from the manuscript.
Finally, I would have some clarifications and questions:
- I would like to know what is the overall prevalence of children/adolescents with blood pressure above the 90th percentile.
- I would like to know what is the prevalence of overweight or obese individuals, possibly using IOTF criteria (Cole BMJ 2000) and possibly the correlation of this condition with breastfeeding. A summary table with prevalence of blood pressure values above the 90th percentile and weight status would be appropriate.
- How were non-breastfed infants fed? Cow's milk, formulated milk, other?
- What was the mode and timing of introduction of complementary feeding? When was adult feeding introduced? If it is not possible to know for the population under study at least describe the habits prevalent in China.
- At what age is salt added?
- Low salt intake, especially assessing blood pressure, should be considered among the criteria of proper nutrition along with assessment of vegetable, fruit, meat and SSB intake.
- I would be curious to understand why the results of research conducted 10 years ago are proposed today.
Author Response
“Breastfeeding Duration and High Blood Pressure in Children and Adolescents: Results from a Cross-Sectional Study of Seven Provinces in China”
Manuscript Number: 1817011
Rebuttal letter
Dear Dr. Lona Li:
Thank you very much for your letter and constructive comments. We have revised the manuscript and would like to re-submit for your consideration. The amendments are highlighted in red in the revised manuscript, and the point by point responses to the reviewer’s comments are listed below.
#Reviewer 2:
The manuscript "Dietary Behaviors May Modify the Association between Breastfeeding Duration and High Blood Pressure in Children and Adolescents: A National Cross-Sectional Study in China" aims to correlate blood pressure values at age 7-18 years with breastfeeding duration and other aspects of feeding behavior in a large population of nearly 30,00 children and adolescents. The results are intriguing because the authors come to the conclusion that prolonging breastfeeding beyond 12 months would have a negative effect on blood pressure, calling into question, at least in this respect, the widely held view that breast milk constitutes some sort of magic potion for the prevention of almost all diseases, including hypertension. The authors themselves do not escape this positive preconception as they state in their conclusions that: "the benefits of breastfeeding might start at the age of 6 months and begin to wane around the age of 12 months" while in the results they show that, compared to non-breastfed, the group of breastfed for 6-12 months have only a modest statistically significant positive effect only on diastolic blood pressure and only in males, while in the group of breastfed only up to 5 months the only significant finding is a small worsening in females. To summarize, in a very large sample, after correcting for numerous confounding factors and having non-breastfed as a reference, those breastfed for more than one year have worse blood pressure values, while those in the other two groups (breastfed only up to 5 months and breastfed for 6-12 months) breast milk would have almost no effect on blood pressure. Relative categorical analysis on subjects with blood pressure values above the 90th percentile (Table 3) also demonstrates a statistically significant correlation (in a worsening sense, i.e., an increase in OR) only in those breastfed for more than one year, while only a nonsignificant trend, positive for males and negative for females, in the other two groups of breastfed. Therefore, I do not think it is correct to speak of a U-shaped curve given the uncertainty of the data of the eventual descending part of the curve.
Response: Thanks for your comments. We rethink the results carefully and believe that a U-shaped curve is not much accurate for the association between breastfeeding duration and childhood BP. According to your suggestion, we revised throughout the manuscript and came to a conclusion as follows:
Based on national representative data, there was no evidence that longer duration of breastfeeding is protective against childhood HBP. Breastfeeding for 6-12 months may be beneficial to BP, while prolonged breastfeeding duration might increase the odds of HBP in children and adolescents. (line 31-34)
The study has a few problems. In particular due to the fact that the percentage of subjects with high blood pressure values (above the 90th percentile) is very high. In theory, only 10% of children with values above the 90th percentile should be expected but, for many reasons, even a relatively large deviation from these values might be acceptable. In the manuscript the deviation appears excessive being at least double, and in some cases, almost triple the expected (Table 3). This may have some explanations that should be discussed and possibly put into the limitations of the study. The AAP guideline 2017 Clinical Practice Guideline for Screening and Management of High Blood Pressure in Children and Adolescents Flynn J. Volume 140, Issue 3 September 2017 gives the following guidance: “if the initial BP is elevated (≥90th percentile), providers should perform 2 additional blood pressure measurements at the same visit and average them. If using auscultation, this averaged measurement is used to determine the child's blood pressure category (normal, elevated blood pressure, stage 1 hypertension, or stage 2 hypertension)”. Thus, having used the average of only two first measurements in each child, it is not possible to confirm the diagnosis of elevated blood pression (or hypertension if the values were above the 95th percentile). Furthermore, it is known that subsequent measurements tend to give lower values than the first ones. Despite this, the prevalence of children/adolescents with elevated values in the sample seems excessive and should be commented on. Doubts may also arise as to how blood pressure is taken.
Response: Thanks for your comments. According to your following comments, we have revised the definition of HBP as BP values above the 95th percentile, based on the fourth report of on the diagnosis, evaluation, and treatment of high blood pressure in children and adolescents.
We rechecked the data and the definition, and we described in the Methods that HBP was defined as if the SBP and/or DBP levels of children and adolescents ≥ the age-, sex- and height-specific 95th percentile references[22] (line 129-131). That was, SBP ≥95th percentiles, DBP ≥95th percentiles, and both SBP and DBP ≥95th percentiles were all defined as HBP in children and adolescents. So the prevalence of HBP was higher than 5% and reached to approximately 10%. The HBP prevalence in the total population, boys and girls was 15.23%, 15.80% and 14.61%, respectively. As you say, for many reasons, a relatively large deviation from these values might be acceptable.
In addition, SBP was defined as the onset of “tapping” Korotkoff sound (K1), and DBP was defined as the fifth Korotkoff sound (K5). BP was measured twice with a 1 min gap between replicates. If the measured difference was >10 mmHg, measurement was repeated until the final two measures differed ≤10 mmHg, and the average of SBP and DBP in the final two measures was used in analyses. However, the BP measurement was collected on one occasion, and we acknowledged that the estimated prevalence of HBP may be overestimated. One meta-analysis underlined the necessity of measuring BP on at least three separate occasions to identify a hypertensive child in clinical practice or to accurately estimate the true prevalence of hypertension in a pediatric population, when compared with visit 1, the prevalence of HBP decreased by 53.7% during visit 2 and by 77.7% during visit 3[1].
To be noted, the WHO recommends using the average of three BP readings at one visit in risk factor surveys[2], and a cross-sectional study including 5207 children and adolescents 10-14 years of age in Switzerland found that the second value of two measurements at a visit might be sufficient for screening elevated BP[3].
Therefore, we have improved the description in the Methods and added this points as one limitation in the revised manuscript as follows:
Methods: Systolic blood pressure (SBP) was defined as the onset of “tapping” Korotkoff sound (K1), and diastolic blood pressure (DBP) was defined as the fifth Korotkoff sound (K5). BP was measured twice with a 1 min gap between replicates. If the measured difference was >10 mmHg, measurement was repeated until the final two measures differed ≤10 mmHg, and the average of SBP and DBP in the final two measures was used in analyses. (line 117-122)
limitation: Firstly, the estimated prevalence of HBP may be overestimated since the BP measurement was collected on one occasion[45]. In the future study, BP measurements with multiple visits or using ambulatory BP monitor would be better. (line 338-340)
Reference:
- Sun J, Steffen LM, Ma C, Liang Y, Xi B. Definition of pediatric hypertension: are blood pressure measurements on three separate occasions necessary? Hypertens Res. 2017 May;40(5):496-503.
- World Health Organization. WHO STEPS Surveillance Manual. Geneva: WHO; 2008.
- Outdili Z, Marti-Soler H, Bovet P, Chiolero A. Performance of blood pressure measurements at an initial screening visit for the diagnosis of hypertension in children. J Clin Hypertens. 2019;21:1352-
With respect to the choice of cuff to be used in each child, "The fourth report on the diagnosis, evaluation, and treatment of high blood pressure in children and adolescents. Pediatrics. 2004, 114, 555-576" recommends, "by convention, an appropriate cuff size is a cuff with an inflatable bladder width that is at least 40 percent of the arm circumference at a point midway between the olecranon and the acromion," whereas the manuscript methods mention 50-75 percent of the arm circumference (line 137) which would suggest the use of cuffs that are too large in many cases, but this would have resulted in an underestimation of the actual blood pressure values. Conversely, in severely overweight adolescents it would have been necessary to use cuffs larger than those available and, in this case, the values would be overestimated. Finally, age is not a criterion for selecting the headset (line136). Also, for proper interpretation of pressure values, it would be more correct to parameterize the stature percentile to US standards and not to Chinese height standards (line 153), since US pressure percentiles are used. But these latter are marginal observations. Despite these observations about the suboptimal accuracy of the measurements, as pressure was measured the same way for everyone, the interesting finding of the association between prolonged breastfeeding and increased blood pressure values at 7-18 years remains.
Response: Thanks for your good comments! We agreed that an appropriate cuff size is a cuff with an inflatable bladder width that is at least 40 percent of the arm circumference at a point midway between the olecranon and the acromion, but since the original version of manuscript did not stratify the population based on BMI Z scores, therefore our manuscript methods mentioned 50-75 percent of the arm circumference including both normal weight or severely overweight children and adolescents. Also, we delete the determinant of age since age is not a criterion for selecting the headset. According to your suggestion, we have revised the Methods as follows:
Methods: Three cuff sizes (7, 9, and 12 cm width) were selected according to the mid-upper arm circumference of the children, which stipulated that the cuff bladder width should cover 40% of the mid-arm circumference. In severely overweight adolescents it would have been necessary to use cuffs larger than those available. (line 112-115)
In addition, thanks for your reminder, we have used the US standards and revised the sentences as follows:
Age-, sex-, and height-specific BP Z score was calculated according to the reference range of the National High Blood Pressure Education Program Working Group on High Blood Pressure in Children and Adolescents[22]. HBP was defined as if the SBP and/or DBP levels of children and adolescents ≥ the age-, sex- and height-specific 95th percentile references[22]. (line 127-131)
Also, we have updated all the results based on the US standards in the Tables, the results did not change essentially.
The part of the manuscript concerning the effects on blood pressure of more or less healthy dietary behaviors also seems to me unsatisfactory in both writing and content. In lines 31-32 of the abstract the statement that "the risk for high blood pressure brought by prolonged breastfeeding decreased in boys (OR=1.29, 95% CI: 0.83, 2.00)" is incomprehensible without reference to the baseline value of non-breastfed children. The same goes for line 285 where "more pronuonced" should have a comparison term. The comments to Figure 2 (which should be called a table) for the part about boys are acceptable. However, it is puzzling that there is no difference between boys having 1,2 or > 3 healty dietary behaviors. While in females the results have such a random pattern that it suggests that the data were not properly collected. In fact, it is a paradox that girls who have more healty dietary behaviors would have a higher risk of having high blood pressure values. The results of this part of the study are so uncertain that it would be preferable not to cite them in the abstract, or even delete them from the manuscript.
Response: Thanks for your constructive comment!
We agreed with you that the results of this part of the study are uncertain since people experience major fluctuations in dietary behaviors during 7-18 year-old, and the dietary data recall of 7 days might not represent long-term dietary behaviors.
After carefully consideration, we decided to delete the part of the modified effects of dietary behaviors in the manuscript, and focused on the topic of breastfeeding and childhood BP, BP Z scores and odds of HBP both in boys and girls, and we added stratified analyses to test the robustness of the results. The revised title is as follows:
“Breastfeeding Duration and High Blood Pressure in Children and Adolescents: Results from a Cross-Sectional Study of Seven Provinces in China”
To date, the evidence in developing settings such as China is still scarce. In view of the current unsatisfactory situation that the peak of HBP rates is trending increasingly towards younger ages in China, and the previous limited but inconsistent evidence mainly focused on a single area with different standard of living, social infrastructure, and postnatal characteristics, there is an urgent need of effective assessment of national representative data to estimate the effects of duration of breastfeeding on BP. So our research is still valuable for this field.
To fill this gap, based on the representative data from seven provinces or cities in China, the main objective of the present study was to investigate the associations between breastfeeding duration with BP levels, BP Z scores and the odds of HBP in children and adolescents.
Finally, I would have some clarifications and questions:
- I would like to know what is the overall prevalence of children/adolescents with blood pressure above the 90th percentile.
Response: Thanks for your comments. We have updated the results based on the US standards: HBP was defined as if the SBP and/or DBP levels of children and adolescents ≥ the age-, sex- and height-specific 95th percentile references. The overall prevalence of children and adolescents with BP above the 95th percentile was 15.23%. The prevalence in boys and girls was 15.80% and 14.61%, respectively (Table 3). We have added a summary table in Supplementary Table 4 to show the prevalence of BP values above the 95th percentile according to different subgroups.
- I would like to know what is the prevalence of overweight or obese individuals, possibly using IOTF criteria (Cole BMJ 2000) and possibly the correlation of this condition with breastfeeding. A summary table with prevalence of blood pressure values above the 90th percentile and weight status would be appropriate.
Response: Thanks for your valuable comments. IOTF BMI cut-off corresponds to a BMI of 25 kg/m2 (greater than 90.5th and 89.3rd percentile in boys and girls) and 30 kg/m2 (greater than 98.9th and 98.6th percentile in boys and girls) at the age of 18 years defining overweight and obesity, respectively. We have added it in the Methods and added a summary table in Supplementary Table 4 to show the prevalence of BP values above the 95th percentile according to different subgroups. In addition, we added the prevalence of overweight or obesity using IOTF criteria by breastfeeding duration groups in the revised Table 1.
Methods: Overweight and obesity were classified using the sex- and age-specific BMI reference values developed by the International Obesity Task Force (IOTF)[21]. (line 107-109)
Also, we provided a Supplementary Table 1 to show the prevalence of overweight/obesity and the correlation of overweight or obesity with breastfeeding duration, and found a similar trend with HBP:
Therefore, we regarded the covariate of overweight/obesity as a stratified indicator in the main results, we focused on the breastfeeding duration with the BP, stratified by sex, age, overweight/obesity status, birth weight, single-child status, residence area and parental educational attainment, the revised the results for BP, BP Z scores and HBP were presented in Supplementary Table 3 and Figure 1.
(Please see the attached PDF file for the revised Table and Figure)
- How were non-breastfed infants fed? Cow's milk, formulated milk, other?
Response: Yes, thanks for your question. During the survey, we asked the question that “Is your child breastfed?” If parents answered “yes”, We would continue to ask about the duration of breastfeeding. Otherwise, we included all non-breastfed feeding type as non-breastfeeding, including cow's milk, goat's milk or formulated milk, etc. We mentioned this point in the Methods as follows:
The non-breastfeeding type included cow's milk, goat's milk or formulated milk, etc. (line 146-147)
- What was the mode and timing of introduction of complementary feeding? When was adult feeding introduced? If it is not possible to know for the population under study at least describe the habits prevalent in China.
Response: Thanks for your valuable suggestion. We acknowledged that the information of complementary feeding and adult feedings was not available in the present study due to the limited data.
As recommended by WHO[1], timely and adequate complementary food in infants and young children should be introduced between the ages of around 6 months and 2 years old. To promote the infants and children complementary feeding recommendations of WHO, the Ministry of Health of China has developed programs to improve nutrition practices[2], from 2005 to 2015, the age at which a variety of complementary food was introduced as around 6 months in China[3]. Guiding principles for feeding breastfed and non-breastfed children recommend that children aged 6-23 months be fed a variety of foods to ensure that nutrient needs are met. Children 6-23 months of age who consumed foods and beverages from at least five out of eight defined food groups each day. Generally, the progression from liquid food to a mixed diet of family foods occurs in the period from 6 to 24 months of age approximately. Six to eight months of age is the ideal stage to initiate complementary food introduction. Infants begin to become familiar with the taste of food during this stage. At the age of 12 to 23 months, most children have tasted various daily family foods, transitioned to eating with their families, and gradually formed their own eating patterns[4].
According to your suggestion, we have revised the Discussion and added these points as follows:
Also, one study in China suggested that complementary foods should be introduced at around 6 months of age, taking infant developmental readiness into account[40]. Generally, After the addition of the complementary food, the progression from liquid food to family foods occurs in the period from 6 to 24 months of age. During this period, children should consume foods and beverages from at least five out of eight defined food groups each day. Therefore, the type and duration of postnatal feeding may also be a driver of infant growth and a modifiable determinant of cardiovascular health. (line 307-314)
Reference:
- World Health Organization . Complementay Feeding: Report of the Global Consultation, and Summary of Guiding Principles for Complementary Feeding of the Breastfed Child. Geneva: WHO; (2002).
- World Health Organization . U. Infant Young Child Feeding Counselling: An Integrated Course. Geneva: World Health Organization; (2006).
- Liu J, Huo J, Sun J, Huang J, Gong W, Wang O. Prevalence of Complementary Feeding Indicators and Associated Factors Among 6- to 23-Month Breastfed Infants and Young Children in Poor Rural Areas of China. Front Public Health. 2021 Oct 1;9:691894.
- Wang S, Mei Y, Ma ZH, Zhao WH, Tang XJ, Pang XH, et al. The Patterns of Complementary Feeding and Growth among 12 to 23 Month-Old Children in China. Biomed Environ Sci. 2021, 34(11):847-858.
- At what age is salt added?
Response: Thanks. The main taste of Chinese food is salty; accordingly, Chinese people consume a considerable amount of sodium, amounting to more than twice the recommended consumption set by WHO. In China, the recommendation age of salt intake of infants is over 1 year old, we acknowledged that the salt intake should be considered when assessing blood pressure, however, the questionnaire used in the study included the dietary behaviors but not the information of salt intake of children and adolescents. We agreed with your suggestion, therefore we have added this point of view as one limitations in the Discussion as follows:
Fourthly, although we adjusted for a few potential confounders, other important dietary factors influencing BP, such as salt and macronutrient intakes, were not available, residual confounding was still possible. (line 353-355)
- Low salt intake, especially assessing blood pressure, should be considered among the criteria of proper nutrition along with assessment of vegetable, fruit, meat and SSB intake.
Response: Thanks for your valuable comments. As answered to the previous comments, the questionnaire used in the study included the dietary behaviors but not the information of salt intake of children and adolescents. We have added this point of view as one limitations in the Discussion as follows:
Fourthly, although we adjusted for a few potential confounders, other important dietary factors influencing BP, such as salt and macronutrient intakes, were not available, residual confounding was still possible. (line 353-355)
- I would be curious to understand why the results of research conducted 10 years ago are proposed today.
Response: Thanks for your meaningful and interesting comment.
The data was obtained from the baseline of a multi-centered, cluster-controlled trial, aiming to prevent overweight and obesity in children and adolescents from seven provinces or cities of China (Hunan, Ningxia, Tianjin, Chongqing, Liaoning, Shanghai, and Guangzhou; registration number: NCT02343588).
We acknowledged that the results of research were conducted 10 years ago, but as we all know, the process of project proposal drafting, ethical approval, on-site investigation, data inspection, and establishment of an open source database took a long time, and we should also ensure the accuracy of the data as much as possible.
However, we believe that the validation of the scientific questions, especially the scientific topic about breastfeeding and childhood high blood pressure in the current study, is not affected by the time of data collection, at least not so much.
Thanks for your valuable comments, based on your suggestion, the paper is now much improved!

Round 2
Reviewer 2 Report
In this version, the manuscript is much improved.